# Development and Usability Assessment of Virtual Reality- and Haptic Technology-Based Educational Content for Perioperative Nursing Education

**DOI:** 10.3390/healthcare12191947

**Published:** 2024-09-29

**Authors:** Hyeon-Young Kim

**Affiliations:** 1College of Nursing, Sahmyook University, Seoul 01795, Republic of Korea; hyykimm@syu.ac.kr; 2VR Healthcare Content Lab, Seoul 01795, Republic of Korea

**Keywords:** virtual reality, haptic technology, perioperative nursing, education

## Abstract

Background/Objectives: In perioperative nursing practice, nursing students can engage in direct, in-person clinical experiences in perioperative environments; however, they face limitations due to infection and contamination risks. This study aimed to develop and evaluate educational content for perioperative clinical practice for nursing students using virtual reality (VR) and haptic technology. Methods: The program, based on the Unity Engine, was created through programming and followed the system development lifecycle (SDLC) phases of analysis, design, implementation, and evaluation. This program allows nursing students to engage in perioperative practice using VR and haptic technology, overcoming previous environmental limitations and enhancing practical and immersive experiences through multi-sensory stimuli. Results: Expert evaluations indicated that the developed content was deemed suitable for educational use. Additionally, a usability assessment with 29 nursing students revealed high levels of presence, usability, and satisfaction among the participants. Conclusions: This program can serve as a foundation for future research on VR-based perioperative nursing education.

## 1. Introduction

Clinical practice is an essential process for prospective healthcare professionals that enables them to possess basic clinical skills, understand the experiences of patients with diseases, and establish the fundamental attitudes of dedicated medical professionals [1]. Through theoretical knowledge, students acquire the specialized knowledge required to perform the role of a nurse, and through clinical practice, they apply this knowledge to real nursing situations, effectively bridging theory and practice [2]. Additionally, experiencing authentic situations allows students to develop problem-solving skills essential for practical and realistic clinical practice [3]. Therefore, clinical experiences are an integral component for the successful transition from being a student to a licensed nurse.

Working in an operating room (OR) setting requires nurses to possess knowledge and skills unique to the OR, including the management of emergency situations, the operation of specialized equipment, and effective communication with interprofessional team members [4,5]. OR nurses must also be proficient in maintaining a sterile environment and minimizing the risk of hospital-acquired infections [6]. Despite the necessity of this wide range of competencies, pre-licensure nursing programs often face difficulties in providing in-person clinical rotations for students in the OR setting due to the high risk of infections, the need for strict sterility, and concerns about patient privacy [7,8].

To address these challenges, various alternative learning methods such as web-based learning, e-learning, and simulations have been introduced. However, these methods may not fully simulate the level of realism and hands-on experience provided by actual field practice [9]. Jeffries [10] emphasizes the importance of designing simulations in nursing education that closely align with real-life clinical experiences to effectively bridge the gap between theoretical knowledge and practical skills. In her structured framework, she highlights the need to integrate educational outcomes with realistic simulations, ensuring that students and educators benefit from both immediate feedback and structured learning environments.

Interest in the application of information technology (IT) in the healthcare industry continues to grow [11]. In particular, virtual reality (VR), a key technology of the Fourth Industrial Revolution, offers immersive and realistic experiences that enhance learning outcomes and improve the quality of healthcare education [12]. Previous studies have reported that VR and haptic technology contribute to increasing learner immersion and learning effectiveness [13]. Currently, VR simulations that can closely simulate actual human organs are being utilized, thereby saving human cadavers, organs, and medical equipment for the education of prospective medical professionals and residents. VR has also been applied in various fields, such as surgery, psychiatry, and rehabilitation. Notably, VR using haptic technology has been effective in procedures such as catheter insertion, Veress needle insertion, and laparoscopic surgery, and related research has been actively conducted both domestically and internationally [14,15,16]. VR education that incorporates haptic technology, which adds tactile sensations to visual sensations, provides immediate proprioceptive feedback to users, and offers a more realistic simulation, enhancing learners’ confidence and performance capabilities [17].

The aim of this study was to determine the effectiveness of VR and haptic technology-based educational content for OR clinical field practice, specifically designed for nursing students. The focus was on providing an immersive learning experience that simulates real OR scenarios, enhancing the practical skills and knowledge of students in a controlled environment. The specific research objectives are as follows:To analyze the literature and investigate the current status and user needs for VR and haptic technology-based OR clinical field education for prospective healthcare professionals;To develop VR and haptic technology-based OR clinical field education content for experiential learning by prospective healthcare professionals;To evaluate the efficacy of the developed VR and haptic technology-based OR clinical field education content through expert assessments and usability evaluations by prospective healthcare professionals.

## 2. Materials and Methods

### 2.1. Research Design and Setting

This study employed a research and development (R&D) method to develop a VR and haptic technology-based perioperative nursing practice educational content program using Unity Engine. The perioperative nursing education content was developed based on the System Development Life Cycle (SDLC), which follows four key phases: analysis, design, implementation, and evaluation [18] (Figure 1). This structured approach ensured a systematic process to meet the educational needs of nursing students.

### 2.2. Methodology

#### 2.2.1. Phase 1: Analysis

To develop an educational program that accurately reflects the needs of practical fieldwork, we conducted a comprehensive literature search focusing on key areas such as VR, health personnel, surgery, perioperative nursing, education, and patient simulation. The literature search was performed across multiple databases, including PubMed, Embase, CINAHL, and the Cochrane Library, between March 2020 and April 2020. The search was restricted to articles published between January 2000 and April 2020 to capture key developments in these fields over the last two decades. The search strategy was constructed using a combination of relevant Medical Subject Headings (MeSH) and keywords such as “Students, Nursing”, “Students, Medical”, “Nurses”, “Virtual Reality”, and “Augmented Reality”. These keywords were combined using the Boolean operators “OR” and “AND” to refine the search for relevant studies. We included only studies published in English, focused on educational interventions using VR and related technologies. We also surveyed the demand for VR perioperative nursing content among 29 fourth-year nursing students at S University in Seoul. Students with no prior clinical practice experience in the OR participated voluntarily after understanding the study’s purpose. The survey assessed their experience with VR education, the necessity and usefulness of VR nursing education, and additional content requirements. The insights gathered from the survey were considered during the development of educational content. Furthermore, we selected six perioperative nursing practice topics based on clinical experience and expertise following consultations with four expert nurses, each with over 20 years of experience at a major general hospital in Seoul. This thorough and informed approach ensured that the training program was meticulously designed to meet the needs of clinical practice.

#### 2.2.2. Phase 2: Design

Based on the analysis results, the core content and problem areas were identified, and corresponding content and algorithms were developed to create a comprehensive VR educational program. A scenario was developed through interviews with experts and prospective medical personnel to ensure the inclusion of essential core content that met high-demand areas. This scenario was specifically designed as an OR VR simulation for practical implementation. Suitable application media were selected to ensure efficient delivery, and plans were made to optimize the use of established resources and space. This comprehensive approach, which included detailed reviews and consultations, ensured that the VR educational program was meticulously structured and aligned with the needs of experts and learners. This program was designed to enhance practical skills and knowledge through immersive and interactive learning.

#### 2.2.3. Phase 3: Implementation

The program was created using the Unity 3D engine as the development platform. To link the VIVE PRO Head Mounted Display (HMD) and the Manus Prime II haptic gloves, a dedicated tracker was attached to recognize hand sensations and whole-body motions from a first-person perspective. The virtual environment included features such as a surgical scrub sink, a Da Vinci surgical robot, an operating bed, a shadowless surgical lights, a Bovie electrosurgical unit, a ventilator, and various kinds of monitors.

The learning content was developed to facilitate training in a VR laboratory, offering an alternative to traditional OR training. This VR setup allowed users to learn about surgical instruments, observe their appearance, understand their use, and experience realistic operations. Through VR, users could interact with and manipulate devices, thereby enhancing the realism of their training experience. In addition, the VR environment included auditory elements such as guidance explanations, background music, and sound effects to further immerse the user. The program was developed in collaboration with a company specializing in VR content production based on the designed content and algorithm. This comprehensive approach ensured that the VR educational program was well structured and aligned with the needs of both experts and learners.

#### 2.2.4. Phase 4: Evaluation

The content and platform developed were thoroughly evaluated by a team of four experts comprising an education team leader with over 30 years of OR clinical experience who had participated in information system development, a computer engineering professor, and two VR content developers. An expert evaluation was conducted by considering the VR content and user interface (UI) design area, along with a heuristic assessment. A heuristic evaluation based on nine principles provided a comprehensive assessment of usability and functionality.

After applying the content, we conducted evaluations with 29 nursing students to assess its acceptance and usability. These evaluations focused on the presence and usability of and satisfaction with the VR experience. Data collection for expert evaluation and usability assessment for prospective medical personnel use began on 17 August 2022. The participants experienced the program developed in the VR Healthcare Content Lab at S University in Seoul until 19 August. Data were collected through surveys. We followed a rigorous evaluation process to assess the efficacy, engagement, and user-friendliness of our VR training programs.

### 2.3. Research Instruments

#### 2.3.1. Heuristic Evaluation

The VR heuristic evaluation was conducted based on ten principles of heuristic evaluation [19] and nine principles of VR heuristics developed by Murtza et al. [20] specifically for virtual environment systems. The severity of each principle was assessed on a 5-point scale ranging from 0 (no usability problems) to 4 (usability problems that must be corrected).

#### 2.3.2. VR Content and UI Design Assessment

The VR content and UI design assessment utilized the tool applied by Ko and Jung [21] for the development of the augmented reality surgical nursing practice education app. This tool consists of 9 content-related questions and 11 design-related questions, each rated on a 5-point Likert scale. As indicated by previous studies, the reliability of the instrument showed a Cronbach’s alpha score of 0.83. In this study, Cronbach’s alpha was 0.85.

#### 2.3.3. Presence

To measure the sense of presence in a virtual environment, we used a tool developed by Schubert et al. [22], which was later modified and supplemented by Kang et al. [23]. The reliability of the tool was indicated by a Cronbach’s alpha score of 0.91 in Kang et al. [23]. In the present study, Cronbach’s alpha was 0.93.

#### 2.3.4. Usability

Usability was evaluated using the System Usability Scale (SUS) developed by Brooke [24]. The SUS is a questionnaire consisting of ten items, each rated on a 5-point Likert scale ranging from 1 (‘completely disagree’) to 5 (‘strongly agree’). In a previous study [25], Cronbach’s alpha was reported to be 0.91. In this study, Cronbach’s alpha was 0.81.

#### 2.3.5. Satisfaction

The 20 practice satisfaction items used by Ko and Jung [21] were modified and supplemented to suit the purposes of this study. Each item was rated on a 5-point Likert scale, with higher scores indicating higher satisfaction. In the previous study, Cronbach’s alpha was 0.89. In this study, Cronbach’s alpha was 0.92.

### 2.4. Data Analysis

The collected data were analyzed using the SPSS version 25.0 program. Descriptive statistics were used to calculate the participants’ general characteristics, expert evaluations, and usability evaluation values in terms of percentages, frequencies, averages, and standard deviations.

### 2.5. Ethical Considerations

This study adhered to the principles of human research ethics and received ethical approval from the Institutional Review Board (IRB) of Sahmyook University (Approval No. 2-1040781-A-N-012021112HR). After obtaining approval, we explained the research purpose and procedure to the participants, assured them of their anonymity and confidentiality, and informed them of their right to withdraw at any time without any disadvantage. Participation was voluntary, and written consent was obtained from all participants.

## 3. Results

### 3.1. Phase 1: Analysis

A literature review revealed that VR in healthcare education primarily uses HMD devices, and there is a lack of research on VR education using haptic technology for nursing students [26]. To enhance realism, it was necessary to include tactile feedback in the simulations. The results of the survey indicated that 72.4% of students had no experience with VR-based learning and 27.6% had minimal experience through unofficial platforms. Further, nearly 85% of the respondents agreed on the need to integrate VR into perioperative nursing education and 90% believed that VR education was useful for bridging the gap between theoretical knowledge and practical skills. In total, 87.5% of students expressed the need for interactive simulations of common surgical procedures and requested modules for perioperative nursing protocols related to tasks performed by nurses in an actual OR. Based on the survey results and expert consultations, six main perioperative nursing practice topics were selected and developed into VR modules: preoperative preparation, surgical equipment check, patient positioning for surgery, surgical instrument and gauze count, post-anesthesia urinary catheter insertion, and preoperative time-out.

### 3.2. Phase 2: Design

We first established the necessary equipment and environment to develop the VR haptic content. This process included procuring high-performance HMDs, haptic gloves, and support computer systems and software. Then, we designed and implemented detailed algorithms for each module selected during the analysis phase. In particular, we developed precise physical modeling and haptic feedback algorithms to reproduce tactile feedback accurately. This was performed while maintaining consistency between the modules, ensuring that each module could independently provide an optimal user experience (Figure 2).

### 3.3. Phase 3: Implementation

This study developed VR educational content using Unity Engine. The hardware used included the VIVE PRO HMD and Prime II haptic devices. Participants could engage with the VR environment by launching content on a PC and wearing the HMD utilizing head-tracking technology. Additionally, Manus haptic gloves were worn on both hands, allowing users to independently perform actions such as movements, operations, and selections. The VR space experienced by the users was divided into a changing room, preparation room, and OR. The interior of each space was modeled based on the movement paths and dimensions, and the necessary surgical equipment and supplies were modeled to closely simulate a realistic OR environment using actual photographs. Medical staff models were also created with detailed imaging from various angles, including front, left, right, and 45-degree views. Each module was designed to be completed in approximately five minutes, and users could exit the screen at any time during the session by clicking on the yellow home button placed at the top. If a user failed the learning task, they could use the Reset button to retry, enabling stress-free repetitive learning. After donning the HMD, the arrows guided the user to the tasks that they needed to perform, ensuring ease of operation. The background sounds and medical staff voices from the OR were integrated to enhance realism. Finally, VIVE tracker 3.0 was integrated with Manus haptic gloves to enable detailed hand movements, allowing for individual finger control. This setup not only captured hand sensations but also recognized full-body motion from a first-person perspective, creating an immersive and responsive training experience.

Additionally, participants entered the virtual OR environment after wearing the HMD and haptic gloves. A tutorial guided them through the steps necessary to complete the tasks, such as performing surgical rubbing and gowning before entering the OR. Once the tutorial was completed, users could choose from six different scenarios, selecting the desired module by touching it through the haptic gloves. Each module involved active participation and physical interaction using the gloves.

For example, the patient preparation module required participants to review the patient’s surgical information, check picture archiving communication system (PACS) images, and conduct a preoperative OR inspection. In the equipment verification module, users monitored various patient parameters and operated the virtual electrosurgical unit (ESU), activated shadowless lights, and set surgical timers. The patient positioning module allowed users to adjust the patient’s position according to the type of surgery, including supine, prone, lateral, lithotomy, jackknife, and Trendelenburg positions. In the surgical count module, users counted gauzes, tools, and sharps used during surgery, with automatic recording via tactile contact. The catheter insertion module allowed participants to insert a catheter post-anesthesia while maintaining sterility, with the necessary tools being automatically set up. Lastly, the preoperative time-out module enabled users to experience a real-time simulation of the time-out process using recorded audio, ensuring adherence to OR safety protocols.

This comprehensive VR setup provided an alternative to traditional OR training, allowing participants to familiarize themselves with surgical instruments, observe their usage, and engage in realistic procedures (Figure 3). The program was further enhanced by auditory guidance, background music, and sound effects, ensuring an immersive and dynamic training environment. Developed in collaboration with VR content production experts, the program was designed to meet the educational needs of both learners and healthcare professionals.

### 3.4. Phase 4: Evaluation

#### 3.4.1. Heuristic Evaluation

The heuristic evaluation of the VR haptic content identified issues in four out of nine categories: physical space constraints, glitches, mental comfort, and user interface design. In the order of severity, the most significant issue was related to glitches, where limitations in the haptic experience were noted. Regarding physical space constraints, participants reported a lack of sufficient physical space to move around. Under mental comfort, some participants experienced slight dizziness. Lastly, regarding user interface design, there was feedback suggesting the need for improved interface speed and additional convenient functions (e.g., buttons). Based on the evaluation results, items that received an average severity score of two or higher were prioritized for system modifications. Items with a severity score of one but deemed necessary were also selected for modification. Consequently, ten issues were chosen for content revision, with the specific improvements detailed in Table 1.

#### 3.4.2. VR Content and UI Design Assessment

The assessment of the VR content and UI design yielded an overall average score of 80.5 ± 5.45 out of 100, indicating a generally high level of user satisfaction. This score reflects positive evaluations across various criteria, including usability, understandability, and consistency. The experts particularly highlighted the content’s utility to users, noting its effectiveness in conveying information. Overall, the content received a favorable assessment from experts. The detailed analysis results are listed in Table 2.

#### 3.4.3. Presence

The evaluation of user presence in the VR content yielded an average score of 16.14 ± 3.50 out of 20, indicating high presence. Most users reported feeling as if they were interacting with virtual characters or objects (4.21 ± 0.77) and experienced a sense of the virtual object being part of their reality (4.07 ± 1.00). However, the least favorable response was related to the sensation of a virtual object moving toward them (3.76 ± 1.06) (Table 3).

#### 3.4.4. Usability

The average scores for each item in the SUS are presented in Table 4. Items 7, 9, 1, and 5 were among the highest scoring items, whereas items 2, 8, 4, and 10 were among the lowest scoring items. The average SUS score for evaluating the usability of the content was 74.40, which exceeds the benchmark score of 68 and indicates an above-average usability level (Table 4).

#### 3.4.5. Satisfaction

The evaluation of user satisfaction with the VR content resulted in an average score of 86.97 ± 9.53 out of 100, reflecting a generally high level of satisfaction. Notably, the lowest score was for the item concerning the understanding of practical methods without assistance, which received a rating of 3.59 ± 1.21 (Table 5).

## 4. Discussion

This study developed and evaluated the usability of perioperative nursing educational content that integrates VR and haptic technology. A usability assessment was conducted with 29 nursing students, indicating that the system was feasible for use and had substantial educational value. Most users reported high levels of immersion and satisfaction, suggesting that this technology could be effective in nursing education.

Previous studies have highlighted a gap between the competencies that new graduates possess and the expectations of nurse managers, particularly in high-stakes clinical settings. Key skills such as critical thinking, communication, and professionalism are often rated more highly by nurse managers than by new graduates, underscoring the need for enhanced educational tools [27]. This study contributes to closing that gap by developing immersive perioperative content that focuses on essential skills required in operating room environments, particularly in scenarios that demand precise hand movements and critical decision-making.

Ricca et al. [28] provide further insights into the effectiveness of hand visualization in VR-based training environments. Their study suggests that while users often prefer to see their hands during VR tasks, the presence of hand visualization does not significantly impact performance in motor skill tasks involving tool manipulation. This finding aligns with the current study, which incorporated haptic feedback and precise hand movements to simulate perioperative procedures. While hand visualization could enhance user immersion, it may not be necessary for achieving high performance in motor skill acquisition. Thus, our VR program focuses on both visual and haptic feedback to ensure a realistic and immersive learning environment, particularly for the hands-on skills required in the OR.

The existing VR-based surgical education primarily utilizes HMDs to deliver visual and auditory information [29]. However, this approach offers limited immersion and lacks physical interaction, which can reduce learning effectiveness. In the medical field, realistic feedback is crucial, but existing content has often failed to provide this adequately [30,31].

Haptic technology plays a critical role in overcoming these limitations. Providing multi-sensory stimuli such as touch, force, and movement enhances learners’ practical experiences and simulates real-life practice, thereby improving learning outcomes [29]. Specifically, haptic feedback during the manipulation of surgical instruments can create an environment that closely resembles actual surgical practice by providing physical resistance and pressure [32].

Similar to the educational benefits observed in escape rooms [33], which enhance critical thinking, problem-solving, and teamwork, the VR and haptic technologies used in this study offer an immersive experience that promotes these essential skills in perioperative settings. Both approaches emphasize active participation and teamwork, offering students a safe yet realistic platform to practice crucial clinical skills. Incorporating these diverse methods into nursing education addresses the gap between theoretical knowledge and practical application, ensuring more comprehensive skill development in high-stakes environments.

Previous studies primarily developed educational content for doctors, medical students, and dental students [34,35]. For instance, a dental drilling simulator using a haptic pen and HMD and intravenous needle insertion training using haptic gloves are notable examples. While these studies have shown positive results in enhancing learners’ confidence and accuracy, there is a relative lack of research applying these technologies to nursing education. Our study serves as foundational research aimed at addressing this gap by developing surgical nursing education content tailored to nursing students. By integrating haptic technology, we aimed to provide an immersive and realistic learning experience, particularly in situations where precise hand movements and sensory integration are critical in the OR. This exploratory work lays the groundwork for future studies to expand upon, potentially comparing these methods with traditional or web-based learning systems.

The significance of this study lies in its innovative approach as a foundational step toward addressing the gaps in perioperative nursing education. First, this study develops six unique VR scenarios—preoperative preparation, surgical equipment checks, patient positioning, surgical instrument and gauze counting, post-anesthesia catheter insertion, and preoperative time-out. These scenarios provide practical applications for theoretical knowledge, making the learning process more interactive and engaging. Second, integrating haptic gloves with the VIVE system sets a new educational standard by incorporating tactile feedback. This feature improves skill acquisition, especially for operating room procedures that require precise hand movements. The technology allows for a more immersive and realistic learning experience, making it highly valuable for nursing students who need hands-on practice. Third, this study enhances educational value by providing a safe and controlled environment for repetitive learning. This is crucial, as it reduces the risk of infection or contamination, enabling students to practice essential procedures as many times as needed, a challenge often present in real clinical settings. Lastly, much like the collaborative benefits observed in escape rooms, the VR and haptic technologies used in this study foster teamwork and active participation in perioperative scenarios. Both methods help to bridge the gap between theoretical knowledge and practical application, promoting the comprehensive skill development that is essential in high-stakes perioperative environments.

### Contributions

The main contributions of this study are as follows:Innovative Integration of Haptic Technology in Nursing Education: This study pioneers the integration of haptic feedback into a perioperative nursing education program, offering a multi-sensory learning experience that enhances realism and skill acquisition in operating room procedures;Development of Specialized VR Content for Nursing Students: Unlike prior studies that focus on medical or dental students, this research fills a gap by creating surgical scenarios specifically tailored for nursing students, addressing the competencies required in the nursing field;Addressing Key Competencies in High-Stakes Environments: This work contributes to closing the competency gap between new nursing graduates and nurse managers’ expectations by providing a practical platform for students to develop essential skills such as critical decision-making and precise hand coordination;Pilot Study for Future Research and Practical Application: As one of the first usability assessments of VR and haptic technology-based content in nursing education, this study serves as a foundational piece for future research on the effectiveness of such programs in real-world nursing education.

However, this study has some limitations. First, the small number of participants and the specific research environment may restrict the generalizability of the findings. Second, physical space constraints may limit the interactions that users can experience in a virtual environment. Third, this study relied primarily on quantitative data, resulting in a lack of in-depth qualitative insights.

Future research should aim to address the current limitations by incorporating a broader range of user groups and clinical scenarios. Expanding studies to include diverse populations and varied clinical environments will contribute to creating more inclusive and comprehensive educational content. Additionally, a cohort study could be conducted to follow newly licensed nurses working in ORs. This study could compare two groups: one that received VR-based training during nursing school and another that either had no clinical experience or underwent live in-person OR clinical training. By examining the confidence and competence of these groups, valuable insights could be gained into the effectiveness of VR training in preparing nurses for real-world clinical practice. This approach will contribute to developing more practical and effective educational programs for nursing students.

## 5. Conclusions

The surgical nursing education program developed by integrating VR and haptic technology successfully overcame the environmental limitations that students might face during practical training by providing an immersive learning experience through multi-sensory stimulation. This program can serve as a valuable foundation for developing VR-based educational content in future nursing education. Further research is needed to continuously evaluate its educational efficacy. The positive outcomes observed in this study suggest that the program has the potential to enhance the quality of nursing education and provide students with opportunities to effectively acquire the competencies required in real clinical environments.

## Figures and Tables

**Figure 1 healthcare-12-01947-f001:**
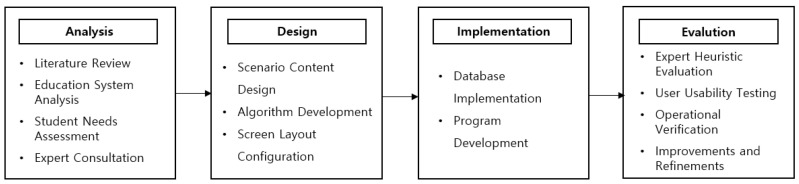
Development process of VR and haptic-based perioperative nursing education content.

**Figure 2 healthcare-12-01947-f002:**
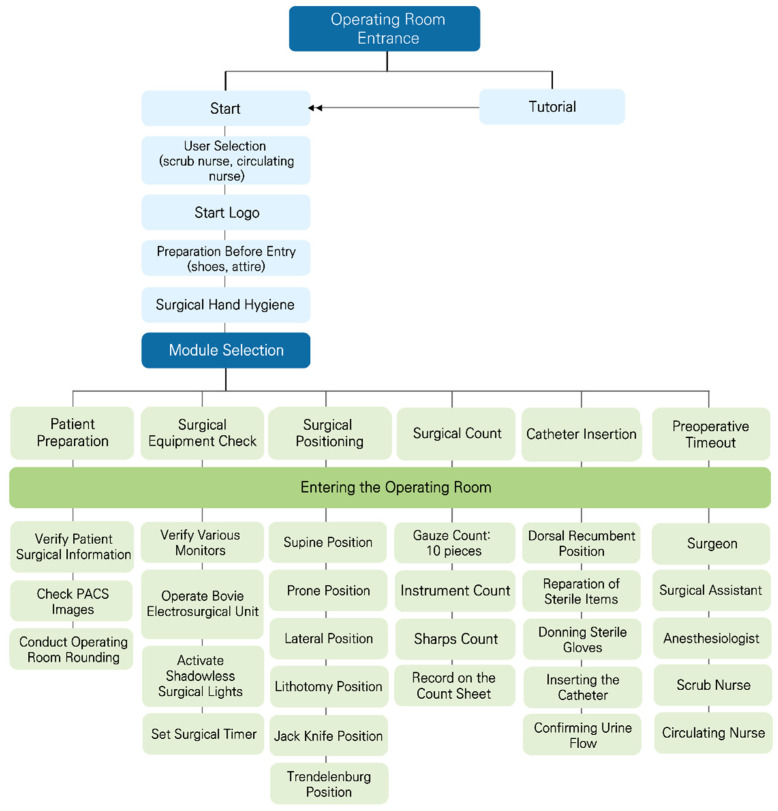
Algorithm for integrating VR-based haptic technology in perioperative nursing training programs.

**Figure 3 healthcare-12-01947-f003:**
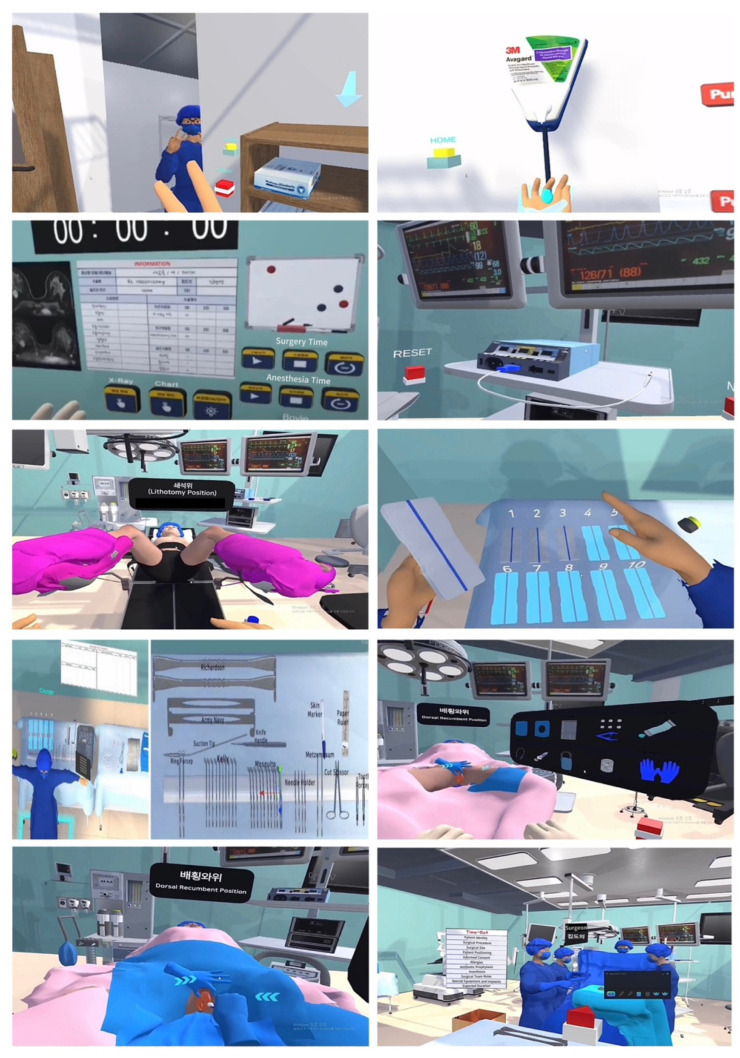
Illustrative screenshots from the VR-based haptic training program for perioperative nursing.

**Table 1 healthcare-12-01947-t001:** List of program improvements.

No	Heuristic	Improvement Content
1	Synchronous Body Movements	-
2	Physical space constraints	Increased the distance between objects and the user during movement
		Changed the movement method from manual adjustments to a teleportation system
3	Immersion	-
4	Glitches	Adjusted the haptic feedback to ensure that only one gauze was pulled at a time during the gauze count process
5	Switch between actual and virtual world	-
6	Cord design	-
7	Headset comfort	-
8	Mental comfort	Adjusted the stereoscopic resolution of the HMD
		Modified the screen display and character movement speed in the HMD
9	User interface design	Redesigned the algorithm for module selection to simplify the process
		Added a Reset button at each location to allow the repetition of tasks
		Changed the color of directional arrows to improve visibility and ease of recognition
		Introduced a menu selection screen to enable users to choose their desired options
		Implemented the surgical count list and correspondingly displayed arrows on the screen, ensuring that the count numbers and guidance voice were accurately synchronized

**Table 2 healthcare-12-01947-t002:** VR content and UI design assessment (N = 4).

Evaluation Factor		No	Questions	Mean ± SD
Contents	Accuracy	1	There is confidence in the healthcare information provided by the app	4.25 ± 0.50
		2	The healthcare information provided by the app is clear	4.50 ± 0.58
	Understanding	3	It is easy to understand healthcare information	4.75 ± 0.50
		4	Health-related terms provided by the app are familiar to the public	4.75 ± 0.50
		5	The level of health care information provided by the app is easy to read	3.75 ± 0.50
	Objectivity	6	Healthcare information is professional information	3.75 ± 0.50
		7	Healthcare information is systematic and specific	4.25 ± 0.50
		8	There is a sign indicating that the information is provided by an authority	3.75 ± 0.50
		9	Health information is provided by medical experts	4.50 ± 0.58
Interface design	Consistency	1	Consistent in color, placement, and presentation	4.75 ± 0.50
		2	The arrangement of icons in the app matches the overall app design	4.75 ± 0.50
		3	Icons in the app are grouped consistently	4.00 ± 0.82
	Compatibility	4	It is easy to understand logically by arranging contents so that they can be accessed sequentially	3.75 ± 0.50
		5	It clearly expresses what the icon means	4.00 ± 0.82
		6	The letters used in the app are in a size and font that can be easily to read by the viewer	4.00 ± 0.82
		7	The visual element works comfortably for the user	3.75 ± 0.50
		8	You can understand the structure of the app at a glance	3.75 ± 0.50
	Vocabulary Accuracy	9	The phrase used in the app is concise	4.50 ± 0.58
		10	The phrase used in the app is correct	4.50 ± 0.58
		11	The phrase used in the app is grammatical	4.75 ± 0.50

**Table 3 healthcare-12-01947-t003:** Presence (N = 29).

No	Questions	Mean ± SD
1	I felt as though a virtual object was moving toward me	3.76 ± 1.06
2	I felt as if a virtual object was right beside me	4.10 ± 1.01
3	I felt that a virtual object was part of my reality	4.07 ± 1.00
4	While participating in the program, I felt like I was interacting with a fictional character or object	4.21 ± 0.77

**Table 4 healthcare-12-01947-t004:** Usability (N = 29).

No	Questions	Mean ± SD
1	I think that I would like to use this system frequently	3.45 ± 0.69
2	I found the system unnecessarily complex	2.10 ± 0.98
3	I thought the system was easy to use	3.31 ± 0.81
4	I think that I would need the support of a technical person to be able to use this system	2.45 ± 0.69
5	I found that the various functions in this system were integrated	3.41 ± 0.73
6	I thought there was too much inconsistency in this system	2.66 ± 0.86
7	I imagine that most people would learn to use this system very quickly	3.86 ± 0.35
8	I found the system very cumbersome to use	2.41 ± 0.78
9	I felt very confident using the system	3.55 ± 0.63
10	I needed to learn a lot of things before I could get going with this system	2.55 ± 0.87

**Table 5 healthcare-12-01947-t005:** Satisfaction (N = 29).

No	Questions	Mean ± SD
1	I was actively engaged in nursing practice	4.72 ± 0.46
2	My interest in practice has increased	4.66 ± 0.48
3	The atmosphere in this practice is positive	4.76 ± 0.44
4	The nursing practice content is logically organized	4.48 ± 0.57
5	The practical textbook is appropriate and useful for learning	4.45 ± 0.63
6	The amount of learning was adequate	4.41 ± 0.73
7	The pace of the training was appropriate	4.48 ± 0.69
8	The content of the practice was engaging	4.83 ± 0.38
9	The practice method was easy to comprehend	4.66 ± 0.61
10	The content of the practice was well structured	4.03 ± 0.98
11	I was able to understand the preparation process before entering the operating room	4.45 ± 0.73
12	I was able to understand the attire required when entering the operating room	4.45 ± 0.63
13	I could understand the procedures involved in applying the correct positioning for surgical patients	3.86 ± 1.25
14	I was able to understand the surgical count procedure	4.41 ± 0.68
15	I am familiar with the surgical timeout process essential for patient safety	4.38 ± 0.73
16	The current practice method was understood without assistance	3.59 ± 1.21
17	The goals set for the practice were successfully achieved	4.03 ± 0.73
18	I was able to acquire the skills needed to perform nursing in an actual operating room	3.86 ± 0.88
19	I gained new knowledge	4.10 ± 0.86
20	I am satisfied with the current practice method	4.38 ± 0.62

## Data Availability

Data are available upon request.

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
