# Peer review of "Development and Usability Assessment of Virtual Reality- and Haptic Technology-Based Educational Content for Perioperative Nursing Education"

_healthcare, 2024, doi:10.3390/healthcare12191947_

Round 1
Reviewer 1 Report
Comments and Suggestions for Authors
The article addresses a real problem in relation to the training of operating theatre nurses. The use of VR with haptic technology seems to be a promising solution to address these limitations of access by student nurses to operating theatres.
Only some text corrections are added in the PDF.

Author Response
Thank you for your insightful feedback on my manuscript. I appreciate your comments, which have guided us in refining our work. The revisions made in the manuscript are indicated in red text for clarity. Below, I address the specific points raised:
Comments and Responses
Comments 1: Please clarify the acronyms
Response 1:
Thank you for your valuable feedback and for giving me the opportunity to improve my manuscript. I have carefully considered your suggestions and made the following revisions to address your comments.
In the sentence: "Interest in the application of information technology (IT) in the healthcare industry continues to grow [11]. In particular, virtual reality (VR), a key technology of the Fourth Industrial Revolution, offers immersive and realistic experiences that enhance learning outcomes and improve the quality of healthcare education [12]." the abbreviations "IT" and "VR" have been defined at their first mention, as per your suggestion to clarify the acronyms. Additionally, the sentence structure has been refined for better flow and readability.
Comments 2: falta bibliografia
Response 2:
Thank you for pointing this out. I have made the following revisions to address the issue of missing references.
The sentence: "The perioperative nursing education content was developed based on the System Development Life Cycle (SDLC), which follows four key phases: analysis, design, implementation, and evaluation [18] (Figure 1). This structured approach ensured a systematic process to meet the educational needs of nursing students." has been revised to include a reference to Langer, A. M., & Langer, A. M. (2008). System development life cycle (SDLC). Analysis and Design of Information Systems: Third Edition, 10-20 to support the use of the SDLC methodology in the development of our educational content. This revision adds a solid foundation for the methodology used in the study.
Reference[18]: Langer, A.M.; Langer, A.M. System development life cycle (SDLC), 3nd ed.; Analysis and Design of Information Systems: New York, USA, 2008, 10-20.
I believe that these revisions address your concerns and have improved the overall quality and depth of the manuscript. Once again, I appreciate your insightful feedback and hope that the changes made meet your expectations.
Thank you for your time and consideration.

Reviewer 2 Report
Comments and Suggestions for Authors
Abstract:
Suggest changing the first sentence to something that clearly states you are referring to in person clinical experiences, rather than 'realistically experiencing the perioperative environment". I first read this to mean that students could realistically experience this environment in a SIM lab etc.
Introduction:
Line 29-31 suggest changing 'real situations' to authentic situations
Line 31 suggest changing to 'Therefore, clinical experiences are an integral component for the successful transition from student to licensed nurse.'
Line 30-44 This whole paragraph seems to contain 2 separate ideas. Could be summarized in the following sentences: Working in an OR setting requires nurses who possess knowledge unique to the OR, in addition to knowledge and competence in areas such as managing emergency situations, operating specialized equipment, and effectively communicating with a multitude of interprofessional team members. Despite this need for a wide range of skills, pre-licensure nursing programs may experience difficulties offering in person clinical rotations for student nurses in this setting, due to the higher risk of hospital acquired infections, need for sterility in the OR setting, and potential challenges regarding patient privacy. To address these issues, various learning methods.....
Suggest rephrasing the aim and removing 'overcome the environmental limitations of clinical practice'. Overcoming the environmental limitations makes it sound as if you sought a way to have the clinical rotations in person in the OR despite the challenges of the environment. Suggest changing to 'The aim of this study was to determine the effectiveness of VR and haptic technology based OR clinical field education content." Remove the objectives as these read as your methods.
On page 12 move the first 2 sentences up to the last paragraph on the previous page
Paragraph on line 336-338 be sure this paragraph has at least 3 sentences
Future research may also include a cohort study that follows newly licensed nurses working in the OR with one cohort who had VR training in nursing school and the other who had either no clinical experience or a live in person OR clinical experience and compare confidence and competence.
Author Response
Thank you for your insightful feedback on my manuscript. I appreciate your comments, which have guided us in refining our work. The revisions made in the manuscript are indicated in blue text for clarity. Below, I address the specific points raised:
Comments and Responses
Comments 1: Abstract: Suggest changing the first sentence to something that clearly states you are referring to in-person clinical experiences, rather than 'realistically experiencing the perioperative environment'.
Response 1: Thank you for your insightful feedback. I have revised the first sentence of the abstract to explicitly refer to "in-person clinical experiences" instead of using the vague phrase "realistically experience the perioperative environment." The updated sentence now reads: "In perioperative nursing practice, nursing students can engage in direct, in-person clinical experiences in perioperative environments; however, they face limitations due to infection and contamination risks." This change ensures clarity and avoids ambiguity by specifying hands-on clinical experiences in real settings rather than simulations or controlled environments.
Comments 2: Line 29-31: Suggest changing 'real situations' to 'authentic situations'.
Response 2: I have adopted your suggestion to replace "real situations" with "authentic situations" to better reflect the nature of the experiences referenced. The revised sentence is: "Additionally, experiencing authentic situations allows students to develop problem-solving skills essential for practical and realistic clinical practice." This change emphasizes that the experiences are genuine and applicable to real-world nursing contexts.
Comments 3: Line 31: Suggest changing to 'Therefore, clinical experiences are an integral component for the successful transition from student to licensed nurse'.
Response 3: Following your recommendation, I have revised the sentence to emphasize the importance of clinical experiences in the transition process. The updated sentence is: "Therefore, clinical experiences are an integral component for the successful transition from student to licensed nurse." This revision underscores the vital role of clinical experiences in both learning and transitioning to full licensure.
Comments 4: Line 30-44: This paragraph contains two separate ideas. It could be summarized as follows: "Working in an OR setting requires nurses who possess knowledge unique to the OR, in addition to managing emergencies, operating specialized equipment, and collaborating with interprofessional teams. Despite this need, pre-licensure nursing programs face difficulties offering in-person clinical rotations due to risks such as infections, sterility requirements, and patient privacy concerns. Various learning methods have been introduced to address these issues."
Response 4: Thank you for your detailed suggestion. I have revised the paragraph as follows to separate the two ideas more clearly: "Working in an operating room (OR) setting requires nurses to possess knowledge and skills unique to the OR, including the management of emergency situations, operation of specialized equipment, and effective communication with interprofessional team members [4,5]. OR nurses must also be proficient in maintaining a sterile environment and minimizing the risk of hospital-acquired infections [6]. Despite the necessity of this wide range of competencies, pre-licensure nursing programs often face difficulties in providing in-person clinical rotations for students in the OR setting due to the high risk of infections, the need for strict sterility, and concerns about patient privacy [7,8].
To address these challenges, various alternative learning methods such as web-based learning, e-learning, and simulations have been introduced. However, these methods may not fully simulate the level of realism and hands-on experience provided by actual field practice [9]" This revision clarifies the distinct issues and solutions in a more organized manner.
Comments 5: Suggest rephrasing the aim and removing 'overcome the environmental limitations of clinical practice'. Change to 'The aim of this study was to determine the effectiveness of VR and haptic technology-based OR clinical field education content'.
Response 5: In line with your recommendation, I have rephrased the aim to focus on determining the effectiveness of the VR and haptic technology-based educational content, removing the reference to overcoming environmental limitations. The revised aim is: "The aim of this study was to determine the effectiveness of VR and haptic technology-based educational content for OR clinical field practice, specifically designed for nursing students. The focus was on providing an immersive learning experience simulating real OR scenarios, enhancing the practical skills and knowledge of students in a controlled environment." I also removed any objectives that might resemble methods, ensuring the aim remains clear and concise.
Comments 6: On page 12, move the first two sentences to the last paragraph of the previous page.
Response 6: As per your suggestion, I have moved the first two sentences from page 12 to the last paragraph of the previous page. This adjustment has been made to improve the flow and coherence of the manuscript, as recommended.
omments 7: Paragraph on line 336-338: Ensure this paragraph contains at least three sentences.
Response 7: I have revised the paragraph to include at least three sentences, as requested. The updated paragraph now reads: "Future research should aim to address the current limitations by incorporating a broader range of user groups and clinical scenarios. Expanding studies to include diverse populations and varied clinical environments will contribute to creating more inclusive and comprehensive educational content. Additionally, a cohort study could be conducted to follow newly licensed nurses working in the OR. This study could compare two groups: one that received VR-based training during nursing school and another that either had no clinical experience or underwent live in-person OR clinical training. By examining the confidence and competence of these groups, valuable insights could be gained into the effectiveness of VR training in preparing nurses for real-world clinical practice. This approach will contribute to developing more practical and effective educational programs for nursing students."
Comments 8: Future research may also include a cohort study that follows newly licensed nurses working in the OR with one cohort who had VR training and another with live in-person experience to compare confidence and competence.
Response 8: I have incorporated your suggestion regarding future research. The updated paragraph reflects your recommendation: "Future research should aim to address the current limitations by incorporating a broader range of user groups and clinical scenarios. Expanding studies to include diverse populations and varied clinical environments will contribute to creating more inclusive and comprehensive educational content. Additionally, a cohort study could be conducted to follow newly licensed nurses working in the OR. This study could compare two groups: one that received VR-based training during nursing school and another that either had no clinical experience or underwent live in-person OR clinical training. By examining the confidence and competence of these groups, valuable insights could be gained into the effectiveness of VR training in preparing nurses for real-world clinical practice. This approach will contribute to developing more practical and effective educational programs for nursing students." This addition enhances the manuscript by outlining a clear direction for future research.
I believe that these revisions address your concerns and have improved the overall quality and depth of the manuscript. Once again, I appreciate your insightful feedback and hope that the changes made meet your expectations.
Thank you for your time and consideration.

Reviewer 3 Report
Comments and Suggestions for Authors
Thank you for considering receiving a review for the article.
The article is interesting and fascinating, very important for publication. I recommend it for publication after the slight corrections:
The central idea of ​​the article is the development of an educational program that leverages VR and haptic technology to provide nursing students with an immersive and interactive experience, allowing them to practice perioperative procedures in a controlled and realistic virtual environment. This innovation aims to address the limitations of traditional clinical practice, such as risks of infection, contamination and limited access to operating rooms.
Overall, the paper presents significant advances in nursing education, demonstrating how emerging technologies such as VR and haptics can revolutionize the way practical skills are taught.
The program's success in improving educational outcomes makes it a valuable tool for future nursing education, with the potential to improve the overall quality of health care education.
The article is well structured and provides comprehensive insights into the development process, making it a useful reference for educators and researchers interested in integrating technology into clinical education.
This article contributes valuable knowledge to the field of nursing education by demonstrating the potential of VR and haptic technologies to improve learning outcomes. This is a promising step forward in creating more interactive and effective educational tools for future healthcare professionals.
I would love to add more articles on simulation rooms in nursing education.
Assessment of Differential Perceptions of Core Nursing Competencies between Nurse Managers and Nursing Graduates: A Cross-Sectional Study
Escape rooms in nursing education: An integrative review of their use, outcomes, and barriers to implementation
Comments on the Quality of English Language
Thank you for considering receiving a review for the article.
The article is interesting and fascinating, very important for publication. I recommend it for publication after the slight corrections:
The central idea of ​​the article is the development of an educational program that leverages VR and haptic technology to provide nursing students with an immersive and interactive experience, allowing them to practice perioperative procedures in a controlled and realistic virtual environment. This innovation aims to address the limitations of traditional clinical practice, such as risks of infection, contamination and limited access to operating rooms.
Overall, the paper presents significant advances in nursing education, demonstrating how emerging technologies such as VR and haptics can revolutionize the way practical skills are taught.
The program's success in improving educational outcomes makes it a valuable tool for future nursing education, with the potential to improve the overall quality of health care education.
The article is well structured and provides comprehensive insights into the development process, making it a useful reference for educators and researchers interested in integrating technology into clinical education.
This article contributes valuable knowledge to the field of nursing education by demonstrating the potential of VR and haptic technologies to improve learning outcomes. This is a promising step forward in creating more interactive and effective educational tools for future healthcare professionals.
I would love to add more articles on simulation rooms in nursing education.
Jeffries, P. R. (2005). Designing simulations for nursing education. Annual review of nursing education, 4, 161-177
Assessment of Differential Perceptions of Core Nursing Competencies between Nurse Managers and Nursing Graduates: A Cross-Sectional Study
Escape rooms in nursing education: An integrative review of their use, outcomes, and barriers to implementation
Author Response
Thank you for your insightful feedback on my manuscript. I appreciate your comments, which have guided us in refining our work. The revisions made in the manuscript are indicated in purple text for clarity. Below, I address the specific points raised:
General Overview
The article is well structured and provides comprehensive insights into the development process, making it a useful reference for educators and researchers interested in integrating technology into clinical education.
This article contributes valuable knowledge to the field of nursing education by demonstrating the potential of VR and haptic technologies to improve learning outcomes. This is a promising step forward in creating more interactive and effective educational tools for future healthcare professionals.
Comments and Responses
Comments 1: I would love to add more articles on simulation rooms in nursing education.
Jeffries, P. R. (2005). Designing simulations for nursing education. Annual review of nursing education, 4, 161-177
Assessment of Differential Perceptions of Core Nursing Competencies between Nurse Managers and Nursing Graduates: A Cross-Sectional Study
Escape rooms in nursing education: An integrative review of their use, outcomes, and barriers to implementation
Response 1: <Addition of articles on simulation rooms in nursing education>
As per your suggestion, I have incorporated three relevant articles to enhance the discussion on simulation-based learning in nursing education:
In the introduction, I added a reference to Jeffries (2005), which emphasizes the importance of designing simulations in nursing education that closely align with real-life clinical experiences. This integration helps to better contextualize the significance of using immersive technologies such as VR and haptic feedback in nursing education to bridge the gap between theoretical learning and practical skill development.
Section in the manuscript:
“Jeffries [10] emphasizes the importance of designing simulations in nursing education that closely align with real-life clinical experiences to effectively bridge the gap between theoretical knowledge and practical skills. In her structured framework, she highlights the need to integrate educational outcomes with realistic simulations, ensuring that students and educators both benefit from immediate feedback and structured learning environments.”
In the discussion, I included the article "Assessment of Differential Perceptions of Core Nursing Competencies between Nurse Managers and Nursing Graduates: A Cross-Sectional Study" to address the gap between nurse managers’ expectations and new nursing graduates’ competencies. This addition supports our argument that immersive technologies such as VR and haptic feedback can help narrow this gap by offering more effective educational tools.
Section in the manuscript:
"Previous studies have highlighted a gap between the competencies new graduates possess and the expectations of nurse managers, particularly in high-stakes clinical settings. Key skills such as critical thinking, communication, and professionalism are often rated more highly by nurse managers than by new graduates, underscoring the need for enhanced educational tools [27]. This study contributes to closing that gap by developing immersive perioperative content that focuses on essential skills required in operating room environments, particularly in scenarios that demand precise hand movements and critical decision-making."
Additionally, I have added a reference to the article "Escape rooms in nursing education: An integrative review of their use, outcomes, and barriers to implementation" to draw a parallel between the educational benefits of escape rooms and the use of VR and haptic technology in our study. Both methods emphasize critical thinking, teamwork, and problem-solving skills in a controlled yet realistic environment, making them highly effective for perioperative education.
Section in the manuscript:
"Similar to the educational benefits observed in escape rooms [33], which enhance critical thinking, problem-solving, and teamwork, the VR and haptic technology used in this study offer an immersive experience that promotes these essential skills in perioperative settings. Both approaches emphasize active participation and teamwork, offering students a safe yet realistic platform to practice crucial clinical skills. Incorporating these diverse methods into nursing education addresses the gap between theoretical knowledge and practical application, ensuring more comprehensive skill development in high-stakes environments."
Comments 2: <General Improvements and Revisions>
Response 2: I have also revised sections of the manuscript for better clarity and alignment with your comments. I ensured that the theoretical underpinnings and the innovative aspects of the VR and haptic technology-based educational program are clearly articulated and linked to the gaps in traditional nursing education methods. Furthermore, the discussion now better reflects how our study contributes to the broader field of nursing education by addressing the lack of immersive, hands-on experiences in perioperative training.
Comments 3: <Limitations and Future Directions>
Response 3: I acknowledged the limitations of our study, including the small number of participants and the specific research environment, which may affect the generalizability of the findings. I also addressed the limitations of physical space in virtual environments and the reliance on quantitative data, as noted in the manuscript.
I believe that these revisions address your concerns and have improved the overall quality and depth of the manuscript. Once again, I appreciate your insightful feedback and hope that the changes made meet your expectations.
Thank you for your time and consideration.

Reviewer 4 Report
Comments and Suggestions for Authors
This manuscript focuses on creating and assessing educational content for perioperative nursing practice using virtual reality (VR) and haptic technology. Developed with the Unity Engine, the program was designed by following the system development lifecycle (SDLC) phases, including analysis, design, implementation, and evaluation. The VR and haptic-based program enables nursing students to participate in perioperative practice, addressing previous environmental constraints and enhancing hands-on, immersive experiences through multi-sensory feedback. Expert evaluations was used to confirm the content's suitability for educational purposes. Furthermore, a usability study with nursing students appears to show high levels of presence, usability, and satisfaction. The paper argues that this provides a foundation for future research on VR-based perioperative nursing education.
This manuscript focuses on creating and assessing educational content for perioperative nursing practice using virtual reality (VR) and haptic technology. The VR and haptic-based program enables nursing students to participate in perioperative practice, addressing previous environmental constraints and enhancing hands-on, immersive experiences through multi-sensory feedback. Expert evaluations were used to confirm the content's suitability for educational purposes. Furthermore, a usability study with nursing students appears to show high levels of presence, usability, and satisfaction. The paper argues that this provides a foundation for future research on VR-based perioperative nursing education.
The manuscript in general is organized, written, and presented well, and the language used throughout the manuscript allows for an easy read and understanding. This problem space of nursing practice using virtual reality (VR) and haptic technology is also interesting and continues to draw attention from a multitude of researchers to this day.
However, I have some concerns about the manuscript that need clarification before it can be considered ready for publication.
-> With respect to the Analysis phase (phase 2.2.1 and 3.1), the manuscript does not go into detail about the search methodology used to identify the relevant literature. What were the search terms?, what were the criterion for inclusion?, what dates were search results used from?, why not other search engines (IEEE VR, ISMAR, ACM CHI) especially if VR and haptics are the focus of this work?
-> On a similar vein, I do not think this work presents the simulation and task that users performed using the implemented simulation (using VR and the haptic gloves) in sufficient or satisfactory detail. I only see two sections that briefly talk about the simulation (sections 2.2.3 and section 3.3) along with an image (Figure 3). Figure 2 needs to be discussed in more detail in prose. Currently it is very difficult to understand what exactly a user was tasked with doing in the VR simulation.
-> There was no video of the simulation submitted, making it difficult to understand just what the conditions looked like. The images in the paper are static and don’t give reviewers an idea of what the user experience or the task was like. I would encourage the authors to submit a video demonstrating what the user experience was like for reviewers/users to not only understand, but also use for replication in future studies.
-> The resolution of Figure 2 is somewhat troubling. The images show interfaces with text in them and they are not clearly visible to the readers.
-> I noticed some missing references with respect to the use of VR in training nurses, and haptics. A lot of work has looked into training medical personnel using haptics and in VR
Ricca, A., Chellali, A., & Otmane, S. (2020, November). Influence of hand visualization on tool-based motor skills training in an immersive VR simulator. In 2020 IEEE International Symposium on Mixed and Augmented Reality (ISMAR) (pp. 260-268). IEEE.
-> I don’t necessarily think this work investigates or effectively assesses a phenomenon. Rather the work presents a simple preliminary evaluation of a VR based educational application that can be used by perioperative nursing practice personnel. This needs to be explicitly clarified in the manuscript. Sure, the authors state that they assess content, but the contribution of the work needs to be stated for what it is.
-> More needs to be said of what is the contribution to the field as a result of this work. What does this allow VR system designers, Healthcare researchers, and medical personnel to take away? What are some of the design guidelines as a result of this work? Where is the novelty?
-> line 47: virtual reality is not necessarily an augmented technology. Sure, to the common person, that terminology may be acceptable, but to XR researchers and enthusiasts , the difference between VR and AR is somewhat crucial in the way it is stated. VR provides for a highly immersive experience in an artificially simulated virtual world, while AR registers and augments virtual content onto our view of the real world. I would hence refrain from using the phrase “(VR) is a promising augmented technology…” because it could be confusing to readers.
->line 52: use of the word replicate seems wrong; maybe simulate is the right word
-> lin 294: move this to next page for clarity
-> discussion paragraph starting from line 322: I do not think this work addresses the gap in the literature; the work simply presents a VR simulation and gathers usability and other ratings from users. A true addressing the gap would involve a comparison of learning between a VR condition and a web based learning system or a traditional mediums of such educational material; also, the six different scenarios are not explained in detail.
Addressing these issues will improve the manuscript before it can be considered ready for publication.
Comments on the Quality of English Language
The manuscript in general is organized, written, and presented well, and the language used throughout the manuscript allows for an easy read and understanding.
Author Response
Thank you for your insightful feedback on my manuscript. I appreciate your comments, which have guided us in refining our work. The revisions made in the manuscript are indicated in green text for clarity. Below, I address the specific points raised:
General Overview
The manuscript in general is organized, written, and presented well, and the language used throughout the manuscript allows for an easy read and understanding. This problem space of nursing practice using virtual reality (VR) and haptic technology is also interesting and continues to draw attention from a multitude of researchers to this day.
Comments and Responses
Comments 1: With respect to the Analysis phase (phase 2.2.1 and 3.1), the manuscript does not go into detail about the search methodology used to identify the relevant literature. What were the search terms?, what were the criterion for inclusion?, what dates were search results used from?, why not other search engines (IEEE VR, ISMAR, ACM CHI) especially if VR and haptics are the focus of this work?
Response 1: Thank you for your detailed feedback, especially regarding the search methodology used in the analysis phase. Based on your suggestions, I have revised Section 2.2.1 to provide more specific details on the databases and search terms used. “To develop an educational program that accurately reflects the needs of practical fieldwork, we conducted a comprehensive literature search focusing on key areas such as VR, health personnel, surgery, perioperative nursing, education, and patient simulation. The literature search was performed across multiple databases, including PubMed, Embase, CINAHL, and the Cochrane Library, between March 2020 and April 2020. The search was restricted to articles published between January 2000 and April 2020 to capture key developments in these fields over the last two decades. The search strategy was constructed using a combination of relevant Medical Subject Headings (MeSH) and keywords such as "Students, Nursing," "Students, Medical," "Nurses," "Virtual Reality," and "Augmented Reality." These keywords were combined using Boolean operators "OR" and "AND" to refine the search for relevant studies. We included only studies published in English, focused on educational interventions using VR and related technologies.”
Regarding the additional databases, such as IEEE VR, ISMAR, and ACM CHI, I acknowledge their importance, particularly for VR and haptics-related content. We have included them as part of our future search strategy for expanding research in the field and improving the comprehensiveness of the literature search.
Comments 2: On a similar vein, I do not think this work presents the simulation and task that users performed using the implemented simulation (using VR and the haptic gloves) in sufficient or satisfactory detail. I only see two sections that briefly talk about the simulation (sections 2.2.3 and section 3.3) along with an image (Figure 3). Figure 2 needs to be discussed in more detail in prose. Currently it is very difficult to understand what exactly a user was tasked with doing in the VR simulation.
Response 2: Thank you for your insightful comments regarding the simulation and the tasks users performed. I have now revised Section 3.3 to provide detailed descriptions of each task users performed in the simulation. I have also elaborated on the interactive elements within the simulation, such as the haptic gloves and immersive VR experience, to ensure a clear understanding of what participants were required to do. Additionally, I have expanded on Figure 2 and its relevance to the simulation tasks to offer greater clarity. “Additionally, participants entered the virtual OR environment after wearing the HMD and haptic gloves. A tutorial guided them through the steps necessary to complete the tasks, such as performing surgical rubbing and gowning before entering the OR. Once the tutorial was completed, users could choose from six different scenarios, selecting the desired module by touching it through the haptic gloves. Each module involved active participation and physical interaction using the gloves.
For example, the patient preparation module required participants to review the patient's surgical information, check picture archiving communication system (PACS) images, and conduct a preoperative OR inspection. In the equipment verification module, users monitored various patient parameters and operated the virtual electrosurgical unit (ESU), activated shadowless lights, and set surgical timers. The patient positioning module allowed users to adjust the patient's position according to the type of surgery, including supine, prone, lateral, lithotomy, jackknife, and trendelenburg positions. In the surgical count module, users counted gauzes, tools, and sharps used during surgery, with automatic recording via tactile contact. The catheter insertion module allowed participants to insert a catheter post-anesthesia while maintaining sterility, with the necessary tools being automatically set up. Lastly, the preoperative time-out module enabled users to experience a real-time simulation of the time-out process using recorded audio, ensuring adherence to OR safety protocols.
This comprehensive VR setup provided an alternative to traditional OR training, allowing participants to familiarize themselves with surgical instruments, observe their usage, and engage in realistic procedures. The program was further enhanced by auditory guidance, background music, and sound effects, ensuring an immersive and dynamic training environment. Developed in collaboration with VR content production experts, the program was designed to meet the educational needs of both learners and healthcare professionals.”
Comments 3: There was no video of the simulation submitted, making it difficult to understand just what the conditions looked like. The images in the paper are static and don’t give reviewers an idea of what the user experience or the task was like. I would encourage the authors to submit a video demonstrating what the user experience was like for reviewers/users to not only understand, but also use for replication in future studies.
Response 3: I agree that a video demonstration would enhance the understanding of the user experience during the simulation. To address your concern, I have created a 58-second supplementary video that visually demonstrates the simulation tasks, including the use of haptic gloves and VR. This video provides a clearer view of the conditions in the virtual operating room, allowing for better replication in future studies. I hope this addition meets your expectations.
Comments 4: The resolution of Figure 2 is somewhat troubling. The images show interfaces with text in them and they are not clearly visible to the readers.
Response 4: I appreciate your feedback on the visual clarity of Figure 2. I have taken steps to improve the resolution of the image and adjusted the font size to enhance readability. These changes were made to ensure that the text and interface elements are clear and legible for all readers. Additionally, the original high-resolution image files will be submitted alongside the revised manuscript for your review.
Comments 5: I noticed some missing references with respect to the use of VR in training nurses, and haptics. A lot of work has looked into training medical personnel using haptics and in VR
Ricca, A., Chellali, A., & Otmane, S. (2020, November). Influence of hand visualization on tool-based motor skills training in an immersive VR simulator. In 2020 IEEE International Symposium on Mixed and Augmented Reality (ISMAR) (pp. 260-268). IEEE.
Response 5: Thank you for your insightful comments regarding the need to include additional references on the use of VR and haptics in medical training. In response, I have incorporated the suggested reference from Ricca et al. (2020) into the Discussion section. This reference highlights the influence of hand visualization on motor skill tasks in VR environments, which aligns with our study’s use of haptic technology and visual feedback for perioperative nursing education. By referencing this work, I aim to clarify the relationship between visual immersion and task performance, as well as further substantiate the relevance of haptic technology in enhancing motor skills training. “Ricca et al. [28] provide further insights into the effectiveness of hand visualization in VR-based training environments. Their study suggests that while users often prefer to see their hands during VR tasks, the presence of hand visualization does not significantly impact performance in motor skill tasks involving tool manipulation. This finding aligns with the current study, which incorporated haptic feedback and precise hand movements to simulate perioperative procedures. While hand visualization could enhance user immersion, it may not be necessary for achieving high performance in motor skill acquisition. Thus, our VR program focuses on both visual and haptic feedback to ensure a realistic and immersive learning environment, particularly for hands-on skills required in the OR.” I hope that these revisions adequately address your concerns and strengthen the manuscript.
Comments 6: I don’t necessarily think this work investigates or effectively assesses a phenomenon. Rather the work presents a simple preliminary evaluation of a VR based educational application that can be used by perioperative nursing practice personnel. This needs to be explicitly clarified in the manuscript. Sure, the authors state that they assess content, but the contribution of the work needs to be stated for what it is.
Response 6: Based on your suggestions, I have revised the manuscript to clearly define the contributions and clarify the evaluation of the VR-based educational application. Specifically, I have added a dedicated section titled "Contributions" within the Discussion to explicitly outline the novel contributions of this study. “The main contributions of this study are as follows:
- Innovative Integration of Haptic Technology in Nursing Education: This study pioneers the integration of haptic feedback into a perioperative nursing education program, offering a multi-sensory learning experience that enhances realism and skill acquisition in operating room procedures.
- Development of Specialized VR Content for Nursing Students: Unlike prior studies that focus on medical or dental students, this research fills a gap by creating surgical scenarios specifically tailored for nursing students, addressing the competencies required in the nursing field.
- Addressing Key Competencies in High-Stakes Environments: This work contributes to closing the competency gap between new nursing graduates and nurse managers' expectations by providing a practical platform for students to develop essential skills such as critical decision-making and precise hand coordination.
- Pilot Study for Future Research and Practical Application: As one of the first usability assessments of VR and haptic technology-based content in nursing education, this study serves as a foundational piece for future research on the effectiveness of such programs in real-world nursing education.“
Comments 7: More needs to be said of what is the contribution to the field as a result of this work. What does this allow VR system designers, Healthcare researchers, and medical personnel to take away? What are some of the design guidelines as a result of this work? Where is the novelty?
Response 7: In response to your comment, I have clarified the specific contributions of this study to VR system design, healthcare research, and perioperative nursing education. “The main contributions of this study are as follows:
- Innovative Integration of Haptic Technology in Nursing Education: This study pioneers the integration of haptic feedback into a perioperative nursing education program, offering a multi-sensory learning experience that enhances realism and skill acquisition in operating room procedures.
- Development of Specialized VR Content for Nursing Students: Unlike prior studies that focus on medical or dental students, this research fills a gap by creating surgical scenarios specifically tailored for nursing students, addressing the competencies required in the nursing field.
- Addressing Key Competencies in High-Stakes Environments: This work contributes to closing the competency gap between new nursing graduates and nurse managers' expectations by providing a practical platform for students to develop essential skills such as critical decision-making and precise hand coordination.
- Pilot Study for Future Research and Practical Application: As one of the first usability assessments of VR and haptic technology-based content in nursing education, this study serves as a foundational piece for future research on the effectiveness of such programs in real-world nursing education.”
Comments 8: line 47: virtual reality is not necessarily an augmented technology. Sure, to the common person, that terminology may be acceptable, but to XR researchers and enthusiasts , the difference between VR and AR is somewhat crucial in the way it is stated. VR provides for a highly immersive experience in an artificially simulated virtual world, while AR registers and augments virtual content onto our view of the real world. I would hence refrain from using the phrase “(VR) is a promising augmented technology…” because it could be confusing to readers.
Response 8: In response to your comment regarding the distinction between virtual reality (VR) and augmented reality (AR), I have revised the sentence to ensure clarity for both general readers and XR researchers. Specifically, I removed the term "augmented technology" and emphasized VR as a key technology of the Fourth Industrial Revolution. The revised sentence now reads:
“Interest in the application of information technology (IT) in the healthcare industry continues to grow [11]. In particular, virtual reality (VR), a key technology of the Fourth Industrial Revolution, offers immersive and realistic experiences that enhance learning outcomes and improve the quality of healthcare education [12].”
This revision aligns with the feedback regarding the importance of accurately representing VR and AR as distinct technologies. I appreciate your input, which has strengthened the clarity of the manuscript.
Comments 9: use of the word replicate seems wrong; maybe simulate is the right word
Response 9: I agree that the term "replicate" may cause confusion in this context, as it typically refers to the exact reproduction of something, whereas the intent in this study is to simulate conditions or behaviors. Based on your suggestion, I have revised the wording in the manuscript, replacing "replicate" with "simulate" in both instances. This adjustment better reflects the purpose of the study, which is to simulate environments and conditions rather than replicate them exactly.
Comments 10: lin 294: move this to next page for clarity
Response 10: To address your comment, we have moved the specified content to the following page to enhance the readability and flow of the manuscript.
Comments 11: discussion paragraph starting from line 322: I do not think this work addresses the gap in the literature; the work simply presents a VR simulation and gathers usability and other ratings from users. A true addressing the gap would involve a comparison of learning between a VR condition and a web based learning system or a traditional mediums of such educational material; also, the six different scenarios are not explained in detail.
Response 11: In response to your comment regarding the study's contribution to the literature, I have revised the discussion section to clarify the exploratory nature of our research. “Our study serves as foundational research aimed at addressing this gap by developing surgical nursing education content tailored to nursing students. By integrating haptic technology, we aimed to provide an immersive and realistic learning experience, particularly in situations where precise hand movements and sensory integration are critical in the OR. This exploratory work lays the groundwork for future studies to expand upon, potentially comparing these methods with traditional or web-based learning systems.
The significance of this study lies in its innovative approach as a foundational step toward addressing the gaps in perioperative nursing education.”
Thank you for your valuable feedback regarding the lack of detail in the explanation of the six different scenarios. In response, I have provided additional descriptions in the results section to clarify each scenario's objectives and relevance in the context of perioperative nursing education. “Additionally, participants entered the virtual OR environment after wearing the HMD and haptic gloves. A tutorial guided them through the steps necessary to complete the tasks, such as performing surgical rubbing and gowning before entering the OR. Once the tutorial was completed, users could choose from six different scenarios, selecting the desired module by touching it through the haptic gloves. Each module involved active participation and physical interaction using the gloves.
For example, the patient preparation module required participants to review the patient's surgical information, check picture archiving communication system (PACS) images, and conduct a preoperative OR inspection. In the equipment verification module, users monitored various patient parameters and operated the virtual electrosurgical unit (ESU), activated shadowless lights, and set surgical timers. The patient positioning module allowed users to adjust the patient's position according to the type of surgery, including supine, prone, lateral, lithotomy, jackknife, and trendelenburg positions. In the surgical count module, users counted gauzes, tools, and sharps used during surgery, with automatic recording via tactile contact. The catheter insertion module allowed participants to insert a catheter post-anesthesia while maintaining sterility, with the necessary tools being automatically set up. Lastly, the preoperative time-out module enabled users to experience a real-time simulation of the time-out process using recorded audio, ensuring adherence to OR safety protocols.
This comprehensive VR setup provided an alternative to traditional OR training, allowing participants to familiarize themselves with surgical instruments, observe their usage, and engage in realistic procedures. The program was further enhanced by auditory guidance, background music, and sound effects, ensuring an immersive and dynamic training environment. Developed in collaboration with VR content production experts, the program was designed to meet the educational needs of both learners and healthcare professionals.”
I believe that these revisions address your concerns and have improved the overall quality and depth of the manuscript. Once again, I appreciate your insightful feedback and hope that the changes made meet your expectations.
Thank you for your time and consideration.
Reviewer 5 Report
Comments and Suggestions for Authors
This manuscript centers on the development and evaluation of an educational virtual reality application for perioperative nursing practice. The application leverages haptic technology and allows nursing students to engage in perioperative practice in a hands-on manner through an immersive experience, thereby overcoming previous environmental limitations. The authors also conducted an evaluation, studying the usability, satisfaction and perceived levels of presence associated with the experience.
Here are few aspects the authors should consider addressing:
- While the authors mention details about the virtual experience, I find that the manuscript seems to lack details about what exactly the users had to do. What would be needed here is a detailed description of the tasks that users had to perform. If this was a perioperative nursing simulation that was developed to offer training experience to nurses, I think it would be beneficial and necessary to include these details.
- In section 2.2.1, the authors describe the analysis process. They mention that libraries like PuMmed and CINAHL were searched but do not explicitly list the other libraries that were consulted, the search criteria, the date ranges for articles included, etc. These are important aspects for a manuscript that claims to perform a systematic analysis of the literature.
- It would be nice to be able to see a video demonstration of the simulation. While the authors attempt to provide details in prose, it becomes hard for a reader to understand what the experience was like from the user's perspective without being able see a video demonstration of the same. Along these lines, I would encourage to submit a supplementary video file that showcases the VR experience for readers to understand their work.
- I find this manuscript lacking in terms of communicating what exactly the contribution of this work is. Along these lines, I would encourage the authors to create a section titled contributions where the authors clearly list all the contributions that this work makes. Specifically, I would call for the authors to highlight the novel contributions that this work makes and what the gaps/voids in the literature are that this work exactly fills. While the manuscript seems to talk about the development and evaluation of an educational VR simulation employed to offer perioperative training to nurses, there is a lack of specification of what the novel contributions are.
- Overall, I think this manuscript is interesting but there are a few areas that warrant improvement before it is ready for publication.
Author Response
Thank you for your insightful feedback on my manuscript. I appreciate your comments, which have guided us in refining our work. The revisions made in the manuscript are indicated in pink text for clarity. Below, I address the specific points raised:
General Overview
This manuscript focuses on the development and evaluation of an educational virtual reality application tailored for perioperative nursing practice. By utilizing haptic technology, the application allows nursing students to engage in hands-on training through an immersive experience, overcoming prior environmental limitations. We also evaluated usability, satisfaction, and perceived presence associated with this experience.
Comments and Responses
Comments 1: While the authors mention details about the virtual experience, I find that the manuscript seems to lack details about what exactly the users had to do. What would be needed here is a detailed description of the tasks that users had to perform. If this was a perioperative nursing simulation that was developed to offer training experience to nurses, I think it would be beneficial and necessary to include these details.
Response 1: I have expanded Section 3.3 to include detailed descriptions of the tasks performed during the VR simulation. Key tasks now include preoperative procedures (surgical scrubbing and gowning), patient positioning, catheter insertion, and instrument counting. This expansion provides a comprehensive view of the training experience, addressing the need for clarity. “Additionally, participants entered the virtual OR environment after wearing the HMD and haptic gloves. A tutorial guided them through the steps necessary to complete the tasks, such as performing surgical rubbing and gowning before entering the OR. Once the tutorial was completed, users could choose from six different scenarios, selecting the desired module by touching it through the haptic gloves. Each module involved active participation and physical interaction using the gloves.
For example, the patient preparation module required participants to review the patient's surgical information, check picture archiving communication system (PACS) images, and conduct a preoperative OR inspection. In the equipment verification module, users monitored various patient parameters and operated the virtual electrosurgical unit (ESU), activated shadowless lights, and set surgical timers. The patient positioning module allowed users to adjust the patient's position according to the type of surgery, including supine, prone, lateral, lithotomy, jackknife, and trendelenburg positions. In the surgical count module, users counted gauzes, tools, and sharps used during surgery, with automatic recording via tactile contact. The catheter insertion module allowed participants to insert a catheter post-anesthesia while maintaining sterility, with the necessary tools being automatically set up. Lastly, the preoperative time-out module enabled users to experience a real-time simulation of the time-out process using recorded audio, ensuring adherence to OR safety protocols.
This comprehensive VR setup provided an alternative to traditional OR training, allowing participants to familiarize themselves with surgical instruments, observe their usage, and engage in realistic procedures. The program was further enhanced by auditory guidance, background music, and sound effects, ensuring an immersive and dynamic training environment. Developed in collaboration with VR content production experts, the program was designed to meet the educational needs of both learners and healthcare professionals.“
Comments 2: In section 2.2.1, the authors describe the analysis process. They mention that libraries like PuMmed and CINAHL were searched but do not explicitly list the other libraries that were consulted, the search criteria, the date ranges for articles included, etc. These are important aspects for a manuscript that claims to perform a systematic analysis of the literature.
Response 2: Section 2.2.1 has been revised to explicitly list all consulted databases (PubMed, Embase, CINAHL, Cochrane Library) and clarify the search criteria, including a date range of January 2000 to April 2020, and the specific Medical Subject Headings (MeSH) used. “To develop an educational program that accurately reflects the needs of practical fieldwork, we conducted a comprehensive literature search focusing on key areas such as VR, health personnel, surgery, perioperative nursing, education, and patient simulation. The literature search was performed across multiple databases, including PubMed, Embase, CINAHL, and the Cochrane Library, between March 2020 and April 2020. The search was restricted to articles published between January 2000 and April 2020 to capture key developments in these fields over the last two decades. The search strategy was constructed using a combination of relevant Medical Subject Headings (MeSH) and keywords such as "Students, Nursing," "Students, Medical," "Nurses," "Virtual Reality," and "Augmented Reality." These keywords were combined using Boolean operators "OR" and "AND" to refine the search for relevant studies. We included only studies published in English, focused on educational interventions using VR and related technologies.”
Comments 3: It would be nice to be able to see a video demonstration of the simulation. While the authors attempt to provide details in prose, it becomes hard for a reader to understand what the experience was like from the user's perspective without being able see a video demonstration of the same. Along these lines, I would encourage to submit a supplementary video file that showcases the VR experience for readers to understand their work.
Response 3: I have included a 58-second supplementary video that showcases the simulation tasks from the user's perspective, providing clarity and a more immersive understanding of the experience.
Comments 4: I find this manuscript lacking in terms of communicating what exactly the contribution of this work is. Along these lines, I would encourage the authors to create a section titled contributions where the authors clearly list all the contributions that this work makes. Specifically, I would call for the authors to highlight the novel contributions that this work makes and what the gaps/voids in the literature are that this work exactly fills. While the manuscript seems to talk about the development and evaluation of an educational VR simulation employed to offer perioperative training to nurses, there is a lack of specification of what the novel contributions are.
Response 4: I have added a dedicated section titled "Contributions" to clearly outline the novel contributions of this work. “The main contributions of this study are as follows:
- Innovative Integration of Haptic Technology in Nursing Education: This study pioneers the integration of haptic feedback into a perioperative nursing education program, offering a multi-sensory learning experience that enhances realism and skill acquisition in operating room procedures.
- Development of Specialized VR Content for Nursing Students: Unlike prior studies that focus on medical or dental students, this research fills a gap by creating surgical scenarios specifically tailored for nursing students, addressing the competencies required in the nursing field.
- Addressing Key Competencies in High-Stakes Environments: This work contributes to closing the competency gap between new nursing graduates and nurse managers' expectations by providing a practical platform for students to develop essential skills such as critical decision-making and precise hand coordination.
- Pilot Study for Future Research and Practical Application: As one of the first usability assessments of VR and haptic technology-based content in nursing education, this study serves as a foundational piece for future research on the effectiveness of such programs in real-world nursing education.“
I believe that these revisions address your concerns and have improved the overall quality and depth of the manuscript. Once again, I appreciate your insightful feedback and hope that the changes made meet your expectations.
Thank you for your time and consideration.

Round 2
Reviewer 4 Report
Comments and Suggestions for Authors
edits seem to address most of my previous concerns.